A global phylogenetic analysis in order to determine the host species and geography dependent features present in the evolution of avian H9N2 influenza hemagglutinin

Dalby Andrew R. 1 A.Dalby@westminster.ac.uk
Iqbal Munir 2
1 Faculty of Science and Technology, University of Westminster , Westminster , UK
2 Avian Viral Diseases Programme, The Pirbright Institute, Compton Laboratory , Newbury, Berkshire , UK
Crandall Keith
Electronic publication date: 2014 Oct 30
Publication date: 2014
Volume: 2
Electronic Location ID: e655
Received 2014 Jul 20; Accepted 2014 Oct 15
Copyright: © 2014 Dalby and Iqbal
Copyright year: 2014
Copyright holder: Dalby and Iqbal
License: This is an open access article distributed under the terms of the Creative Commons Attribution License, which permits unrestricted use, distribution, reproduction and adaptation in any medium and for any purpose provided that it is properly attributed. For attribution, the original author(s), title, publication source (PeerJ) and either DOI or URL of the article must be cited.
License URL: https://creativecommons.org/licenses/by/4.0/

Keywords: H9N2, Avian influenza, Hemagglutinin, Phylogenetics, Geography, Host species

Funding: The work presented was not funded by any specific grant funding.

==============================
A complete phylogenetic analysis of all of the H9N2 hemagglutinin sequences that were collected between 1966 and 2012 was carried out in order to build a picture of the geographical and host specific evolution of the hemagglutinin protein. To improve the quality and applicability of the output data the sequences were divided into subsets based upon location and host species.

The phylogenetic analysis of hemagglutinin reveals that the protein has distinct lineages between China and the Middle East, and that wild birds in both regions retain a distinct form of the H9 molecule, from the same lineage as the ancestral hemagglutinin. The results add further evidence to the hypothesis that the current predominant H9N2 hemagglutinin lineage might have originated in Southern China. The study also shows that there are sampling problems that affect the reliability of this and any similar analysis. This raises questions about the surveillance of H9N2 and the need for wider sampling of the virus in the environment.

The results of this analysis are also consistent with a model where hemagglutinin has predominantly evolved by neutral drift punctuated by occasional selection events. These selective events have produced the current pattern of distinct lineages in the Middle East, Korea and China. This interpretation is in agreement with existing studies that have shown that there is widespread intra-country sequence evolution.

Introduction

The spread of avian influenza viruses (AIV) are a major cause of concern for global animal and public health; these viruses are causing enormous economic losses as well as posing a credible threat for pandemic emergence (Sorrell et al., 2009; Xu et al., 2004; Yu et al., 2011). There has been an increase in the monitoring of disease outbreaks in wild and domestic birds, as well as in other potential hosts such as swine and humans, but we still lack a coordinated global surveillance network (Butler, 2012). The Influenza A/H5N1 virus has been the main focus of international monitoring after a series of recent outbreaks, but the emergence of the A/H1N1 pandemic virus “swine flu” in 2009 showed that other subtypes also pose a serious threat to human health (Cao et al., 2009). Experiments have been carried out to determine the exact factors of bird to human transmission and of droplet transmission of H9N2 viruses (Sorrell et al., 2009).

The H9N2 subtype is a variant of AIV usually associated with low pathogenicity. Due to the lower pathogenicity phenotype of this virus, data collection has been very sporadic. There have been outbreaks of H9N2 in flocks of domestic birds resulting in significant economic loss and with high mortality rates of up to 60% reported during the epizootic of 1998–2001 in Iran (Nili & Asasi, 2002). This subtype has also been shown to pass to pigs, ferrets and guinea pigs, as well as to humans in a small number of cases (Butt et al., 2005; Cheng et al., 2011; Lin et al., 2000; Lv et al., 2012; Peiris et al., 1999; Wan et al., 2008; Xu et al., 2004; Yu et al., 2008; Zhang et al., 2009). Antibodies to the virus have also been found in a sero-epidemiological investigation of poultry workers (Pawar et al., 2012; Wang, Fu & Zheng, 2009). These cross species infections show that in the future the virus may present a serious threat to human health.

The co-circulation of H9N2 with other H5N1, H7N3, H1N1 and H3N2 subtypes has resulted in the emergence of novel reassortant viruses (Monne et al., 2013; Peiris et al., 2001; Sun et al., 2011). The reassorted virus has been shown to possess increased virulence (Iqbal et al., 2009; Marshall et al., 2013). The recent emergence of a novel reassortant H7N9 virus containing internal genes from the H9N2 virus is another of novel AIV in birds which has the capability of infecting humans, with fatal consequences. However in the cases of reassortment human to human transmission has not been demonstrated (Watanabe et al., 2013).

There have been a number of recent studies on the evolution of AIV that have incorporated geographical data available from global influenza monitoring (Fusaro et al., 2011; Haase et al., 2010; Lam et al., 2012; Wallace et al., 2007). With the growth in the global monitoring efforts, and the widespread use of cheaper DNA sequencing technology there has been a rapid expansion in the number of available sequences. Previous phylogenetic studies of H9N2 hemagglutinin, have focussed on sequences from a single location (Banks et al., 2000; Butt et al., 2010; Kim et al., 2006; Li et al., 2005; Song, Han & Chen, 2011; Xu et al., 2007a). The largest previous phylogeographical study was that of Fusaro et al. (2011) who surveyed all of the H9N2 viral segments from the Middle East. Fusaro’s study defined eight geographical regions covering the Middle East and used maximum likelihood methods to construct the phylogenetic analysis. Bayesian methods were used to evaluate the geographical clade distribution. This work has recently been extended to take a more detailed look at viral evolution in Israel and to show that there have been successive introductions from neighbouring countries (Davidson et al., 2014).

A large-scale phylogenetic analysis using eight viral gene segments from 571 complete genomic sequences collected between 1966 and 2009 was carried out by Dong and co-workers (2011). Geographical details were not the focus of this study as the aim was to establish the lineage structure and genotypes present in H9N2. That analysis revealed 74 lineages and 98 genotypes when re-assortment is taken into consideration, but they only identified 7 HA variants.

Investigations of the evolution of influenza A H3N2 hemagglutinin using evidence from flu antigen evolution, have shown how the rates of evolution can vary between selection events (epochs) (Koelle et al., 2006; Smith et al., 2004; Thomas & Hertz, 2012; Wolf et al., 2006). Wagner used this study to suggest a reconciliation between the selectionist and neutralist views in a network based model (Wagner, 2008). Wagner’s model also shows that the order in which mutations take place can have an effect on the selection of a group of mutations. In this way random drift is punctuated by selective epochs. Using mathematical models, Bedford, Rambaut & Pascual (2012) have shown that in A/H3N2 sequence evolution is constrained by canalisation. This agrees with Wagner’s hypothesis if there are times when there is limited drift but occasional bursts where the organism escapes the canalisation (Thomas & Hertz, 2012; Wolf et al., 2006).

It is difficult to assess the performance of phylogenetic models. The methods are sensitive to the distance measures used and this is reflected in the number of sites that can be compared and the number of sequences in the study (Felsenstein & Felenstein, 2004). Clades should be monophyletic if they are produced by divergent evolution (Page & Holmes, 1998). If a lineage is geography and host specific then we would expect all of the sequences that form a clade to share the same labels in terms of geography and the species in which they are found. Single base or amino acid changes are ambiguous and could be a product of either divergence or convergence, but larger conserved patterns, especially if they are non-consecutive in the sequence alignments are good indicators of mutations that are responsible for differentiating between clades. These patterns of change can then be examined in terms of their effects on protein structure and function. Ultimately it is the biological function that determines how selection has been responsible for clade differentiation.

The current study presents a comprehensive phylogenetic analysis of the H9N2 HA, that includes sequences from all of the geographical regions where H9N2 has been reported. This investigation shows how host species and geographical distribution have shaped the evolution of distinct lineages of H9 HA. This reveals clades that are both geographically and host species dependent. From these analyses new hypotheses can be generated for more specific events such as migrations or intra-species infections.

Materials and Methods

The complete set of H9N2 hemagglutinin protein sequences were downloaded from the NCBI on the 22nd of May 2013. The search term used for searching the protein database was H9N2 hemagglutinin. The sequences were exported in FASTA format.

The complete set of H9N2 hemagglutinin nucleotide sequences were downloaded from the NCBI on the 9th of September 2013. The search term used was ((H9N2 hemagglutinin) NOT precursor) NOT partial. The sequences were downloaded in FASTA format.

A dataset for the complete set of Korean H9N2 hemagglutinin sequences was downloaded from the NCBI on the 15th of June 2014. The search term used was ((H9N2 hemagglutinin) NOT H5) Korea).

After removal of long truncations (>40 amino-acid residues) as well as a group of sequences that were actually from other AIV subtypes, the final protein dataset contained 2,045 sequences, the final nucleotide dataset contained 1,052 sequences of which the Korean nucleotide subset contained 64 sequences.

The edited protein and nucleotide datasets were then broken into sub-groups based on the sequence annotations, using a short text matching program written in Perl. The data was split into subsets based on geographical location and host species.

The geographical subsets were based on national boundaries except for China, which was divided into its regions. Where a Chinese region had 10 sequences or less then phylogenetic analysis was not carried out. National borders have been identified as barriers to influenza transmission and so these are appropriate geographical subsets (Wallace & Fitch, 2008). A subset was created for the Americas including all of the North American and the single South American (Argentinian) sequence.

Species trees were created for chicken, duck (including mallard), quail, pheasant and swine. In the past it has not been common practice to produce species specific phylogenetic trees but a recent paper by Worobey, Han & Rambaut (2014) also used this approach. There are numerous wild bird species and also some environmental samples, but these are often only represented by single cases and so subsets were not created.

All of the sequence analysis and editing was carried out in MEGA 5.2 on a Windows 8 computer. Both the protein and gene sequences were aligned using Muscle within the MEGA sequence analysis package, using the default parameters (Edgar, 2004; Tamura et al., 2011).

Model evaluation was carried out for the protein dataset within MEGA. This analysis showed that the Jones–Taylor–Thornton model with gamma distributed rates amongst sites was the best model (Table S1) (Jones, Taylor & Thornton, 1992). Phylogenetic trees for the protein sequences were then constructed using Maximum Likelihood within MEGA using the JTT + G substitution model. Each model was tested with 500 bootstraps replicates. To make the calculations tractable the heuristic nearest neighbour interchange was used. The initial tree was created with neighbourhood joining.

Model evaluation for the corresponding nucleotide dataset showed that the Tamura-Nei substitution model with gamma distributed rates amongst sites was the best performing model (Table S2) (Tamura & Nei, 1993). Phylogenetic trees for the gene sequences were constructed using Maximum Likelihood within MEGA using the TN93 + G substitution model. Models were tested with 500 bootstrap replicates. Smaller numbers of sequences in the gene dataset meant that it was possible to calculate the bootstrap values for the chicken trees, which could not be calculated in the protein trees.

The amino acids responsible for clade formation were determined manually using the alignment view within MEGA. This view shows only the amino acid changes relative to the conserved sequence, and so it is a matter of scanning the sequences looking for substitutions that correspond to the different clusters.

All of the trees have been presented as cladograms rather than as phylograms because the study uses a cladistic approach for the analysis. The topology (ordering of the clusters) is the most significant factor in this investigation rather than the distances between groups. The removal of distance data and also the bootstrap data from the figures improves the clarity of the diagrams, but this omitted information is available in Supplemental Information 5 that includes the full phylogenetic analysis.

Results and Discussion

Global nucleotide phylogenetic tree analysis

A complete phylogenetic analysis of the nucleotide dataset was carried out. Computational limits on memory and processor speed make it impossible to carry out a complete analysis on the protein dataset (especially as it contains almost twice as many sequences). A condensed view of the principal clades from the global nucleotide tree can be seen in Fig. 1. One noticeable absence from the global nucleotide tree is the quail sequences. These were omitted during the editing of the sequences to remove truncated sequences where the ends of the sequences were missing. The virus can be broken into three main clusters labelled A, B and the Main Chinese Clade. (The complete tree can be found in Fig. S1.) There is also a small clade (labelled C) that splits from the root of the tree between clades A and B. This clade contains Chinese sequences that were isolated from chickens between 1999 and 2002. This topology agrees with those from the existing literature except that there is some disagreement in the identification of the lineages (Li et al., 2003; Peiris et al., 2001; Perk et al., 2006; Yu et al., 2008). Ji used a lineage and sub-lineage nomenclature that identified four lineages and 2 sub-lineages corresponding to Clade A (lineages 9.1, 9.2 and 9.3), Clade B (sub-lineage 9.4.1) and the main Chinese Clade (sub-lineage 9.4.2). Huang and co-workers used a lineage naming system based on the prototypes Y439, TY WS 66, G1 and Y280 (Huang et al., 2010). Y439, and TY WS 66 (Turkey, Wisconsin 1966) are both in clade A. G1 is in clade B and Y280 is in the main Chinese clade in China 1 1994–2010. The large study by Dong et al. (2011) identified HK/G1/97, BJ/1/94, HK/289/78, HK/AF157/92, KR/96323/96, DE/113/95 and WI/1/66 as prototype sequences for the different HA lineages. The G1 lineage and WI/1/66 lineages are well established and correspond to Clade B and Clade A respectively. BJ/1/94 is in the same clade as Y280, HK/289/78, HK/AF157/92 and DE/113/95 are not in the current dataset because of truncations, but there are sub-clades for Hong Kong duck sequences and Korean chicken sequences that correspond to HK/289/78 and KR/96323/96 respectively. This suggests that it might be possible to divide Clade A into further sub-clades but given the limited number of sequences sampled there is insufficient evidence to be able to carry this out at present.

Figure 1 Phylogenetic overview.

A compressed view of the complete phylogenetic tree with the major clades shown in a condensed format.

Within the Main Chinese clade there are a series of nested sub-clades labelled China 1–9 where the annotations follow the correct date order, with the exception of clade 8. In clade 8 there is a single sequence from a chicken in the Shandong region collected in 1999, which is a probable outlier. All of the other sequences within the clade are from 2003 onwards, which would be consistent with the splitting dates for the other sub-clades. It is possible that this is an example of convergent evolution between recent Chinese sequences and an earlier branching of the phylogenetic tree, but a single sequence is insufficient evidence to corroborate this. An alternative explanation is that this results from sequencing error but this is also unlikely given the large number of bases that would have to be incorrectly identified (for an alignment between the Shandong sequence, G1, G9/Y280, Wisconsin 1966 sequences see Fig. S2). It is possible that this could be a database error, but again this seems unlikely given the provenance and tracking systems within GenBank. In the absence of database error and a clear evolutionary connection between this lone sequence and the rest of the clade does provide evidence for the inadequate sampling of sequences. This nested structure of the Chinese sequences had previously been reported by Song, Han & Chen (2011).

The expanded tree (Fig. S1) shows that the geographical sampling has not been systematic and that it has been carried out in a sporadic and haphazard manner. There are breaks in the dates of sequence annotations in some of the clades that are homogeneous for location. Date gaps in annotations at specific locations are likely to be a result of inadequate sampling rather than reliable evidence for the loss and subsequent migratory return of the clade. The focus on China because of outbreaks of H9N2 within flocks of domestic birds has resulted in a very dense phylogenetic tree for the Chinese sequences, resulting in an artificially Chinese focused distribution to the phylogenetic trees. There is only a single sequence from South America this is the result of a sampling effect rather than reflecting the actual H9N2 distribution.

Clade A (Fig. 2) corresponds to wild bird infections that are distributed world-wide and include examples from North America, South America, Europe and the Far East. This is an important clade because it also contains the original sequence of the hemagglutinin from the H9N2 subtype that was found in a turkey in Wisconsin in 1966. This clade also contains a number of recent sequences from Korea the US and Europe and so this clade remains extant. The topology of this clade disagrees with that from Kim and co-workers who constructed a tree for regions 1-1104 using DNA Star (Kim et al., 2006). The complete sequence for HA is over 1,700 bases, and the method used here is Maximum Likelihood, whereas DNA Star uses the less reliable UPGMA tree generation algorithm within ClustalW. In their tree the recent Korean sequences were placed outside clade A and beyond the Y280 sequences as a distinct clade. The bootstrap values are high for this region of the tree (>99%) and it has geographical consistency with the rest of the clade, and so the topology presented in the current paper is most likely to be correct.

Figure 2 Clade A.

This is the original lineage of H9N2 that was first isolated in a turkey in Wisconsin in 1966.

Clade B (Fig. 3) contains mostly Middle Eastern sequences although there are a few Chinese sequences that form a sub-clade. This clade corresponds to the extended G1 lineage used by Fusaro et al. (2011) and Monne et al. (2013). Iran and Israel are the two countries most strongly represented in this clade. There are also a number of sequences from the Arabian peninsular and a large grouping from the Indian sub-continent.

Figure 3 Clade B.

This is also known as the G1 lineage.

The sub-lineage structure of Clade B shows that the sequences have evolved substantially from the G1 prototype. The G1 prototype sub-clade became extinct in Iran in 2004 and in Israel in 2007. This was replaced by a new sub-lineage (labelled 725 sub-clade in Fig. 3) that appears to have originated in chicken flocks in Iran in 1998, although it is only found in a large number of Iranian samples after 2004. This sub-lineage is similar to that previously identified by Fusaro et al. (2011) and Monne et al. (2013) (labelled cluster C in their papers). There is a sub-clade from the Indian subcontinent that originated in the Punjab/Haryana region in 2003. There is a linking clade than originated in Pakistan in 2004 before spreading back to Iran in 2009. The final sub-clade seems to have originated in the Arabian Peninsular in 2006 and correspond to cluster B from the studies of Fusaro et al. (2011) and Monne et al. (2013). This sub-lineage then spread to Israel and most recently to Egypt. This is in good agreement with the previous studies on the origin of the Egyptian virus (Abdel-Moneim, Afifi & El-Kady, 2012). The phylogenetic tree shows that after the initial Egyptian outbreak in 2010 there has been a marked diversification in the sequences.

Clade C (Fig. 4) is a Chinese clade that has annotated dates between 1999 and 2001. In the phylogenetic analysis this clade falls between Clade A and Clade B. As there are only a small number of sequences within the clade there is no discernable pattern to the distribution of the sequences within the Chinese regions, although it appears to have originated in Guangdong in 1999. As there are no more sequences after those from 2002 this clade can be considered extinct.

Figure 4 Clade C.

This is a small clade found between clades A and B.

The first Chinese clade contains the G9/Y280 lineage (Fig. 5). After the banning of live quail from poultry markets the G1 lineage disappeared from Chinese poultry leaving only the G9/Y280 lineage (Choi et al., 2004). In the study of Li et al. (2005) four lineages were specified G1, TY/WS/66, Y439 and Beijing/1/94, but no sequences from G1 or Y439 were found in their sample. Cong and co-workers (2007) identified two more lineages within this clade based on antigenic studies and nucleotide phylogenetic trees. These are represented by prototype sequences from Shanghai 1998 and a swine genotype. These lineages only cover a small number of the possible sub-clades within this first Chinese Clade (Fig. 5). As no sequences have been sampled from this clade since 2010 it is possible that the G7/Y280 lineage is now extinct, but this hypothesis cannot be confirmed without a longer period of absence of viruses from this clade.

Figure 5 The main Chinese Clade.

This contains the Y280 lineage but this has divided into a series of deeply rooted nested clades.

This study shows a series of new clades that had not been previously identified and that originate in 1997 in Beijing. This is also the origin of the clade labelled China 2. All of the subsequent nested Chinese clades are related to this Beijing sequence. Some sub-clades such as China 4, China 5 and China 6 also appear to have become extinct and so there is good evidence for successive selective sweeps through the viral population.

As discussed previously the individual unusual sequence from 1999 in China 8 sub-clade is difficult to explain. It might suggest that there is considerable sequence diversity within these nested clades, but because of the low number of individuals carrying a particular sequence it might take a long-time for a sequence to become fixed sufficiently within the population to be found by sporadic sampling. If this is true then all of the clades probably have a much earlier origin and there is a long period before first detection. This causes some concern for tracking the evolution of potential pandemic strains, as they might be circulating quite widely before they are detected for the first time.

Clade analysis of the geographical subsets

From the global phylogenetic tree three interesting geographical subsets were identified; the tree for Korea because it is homogeneous to clade A, and the trees from Iran and Israel as they show successive waves of sequence evolution in clade B. The main Chinese clade has many interesting features but these are difficult if not impossible to untangle and coherently explain because of the limitations of sampling within the data, where there are only small numbers from some regions and then large numbers from others.

Both the protein and nucleotide phylogenetic trees for Korea are shown in Fig. 6. There is a fair agreement between the two trees but once again this emphasizes the problems of sampling and the possible effects this can have on phylogenetic reconstruction. From the nucleotide tree it is tempting to define a clade that contains only wild birds and that suddenly developed in 2005, but this clade is not present in the protein tree, and so it cannot be definitively assigned. There are also some differences in the topology surrounding the swine flu case (marked with a red diamond). Previous studies have investigated the effect of vaccination, which initially suppressed the number of cases that were seen in 2008 before an increase in the number of cases in 2009 (Park et al., 2011). From the trees presented here the period following the introduction of vaccination does correspond to a period of diversification. All of the Korean sequences are from clade A, which is the longest circulating lineage and previously it had not shown a wide diversity of sequences. Vaccination is likely to have had an impact on hemagglutinin evolution, which can be seen in the 2009–2010 clade (Lee & Song, 2013).

Figure 6 The Korean nucleotide and amino acid phylogenetic trees.

(A) The nucleotide phylogenetic tree. (B) The amino acid phylogenetic tree.

The protein and nucleotide phylogenetic trees for Israel and Iran are shown in Fig. 7. In the Iranian tree Clade 1 are duck sequences from Clade A. This is why this clade has the deepest origin within the tree, but it does not contain the earliest sequences. This shows that wild birds can introduce other lineages to a geographical region where another lineage currently dominates.

Figure 7 The Iranian and Israeli nucleotide and amino acid phylogenetic trees.

(A) The Iranian nucleotide phylogenetic tree, (B) The Iranian amino acid phylogenetic tree, (C) The Israeli nucleotide phylogenetic tree, (D) The Israeli amino acid phylogenetic tree.

Clades 2 and 5 in the Iranian tree are the oldest clade from the G1-like lineage in these regions. The G1 lineage appears to have originated in Hong Kong in 1997. The initial G1 clade (Clade 2) becomes extinct in 2003 in the nucleotide tree and in 2005 in the protein tree indicating a selective sweep. This event coincides with the loss of sister Clades 1 and 2 in the Israeli protein tree and Clade 1 in the nucleotide tree. These clades were assigned to cluster A by Fusaro et al. (2011) and Monne et al. (2013). The second G1-like clade (clade 5, labelled sub-clade 725 in the global phylogenetic tree, Fusaro/Monne Cluster D) continued to be found in Iran until 2009. This has no equivalent in the Israeli trees, which seem to have inherited a G1-like lineage which originated in the Indian sub-continent and then circulated in the Arabian peninsular (see the global phylogenetic tree, Fig. S1 and Fig. 3, Fusaro and Monne cluster B) before being found in Israel in 2006/2007 (clade 3 in the nucleotide and protein trees) and in Iran in 2009 (clade 4 in the protein and nucleotide trees). A recent paper has shown that there have in fact been several introductions of H9N2 to Israel from Jordan, and the Arabian Peninsula (Davidson et al., 2014). This newly introduced G1-like sub-lineage seems to have had a selective advantage over the existing viral G1-like sub-lineages but this can only be confirmed by future sampling. Subsequently this new sub-lineage has also spread to Egypt from Israel.

Clade analysis of the host specific subsets

The interesting host species subsets are those for ducks, quail and swine. Ducks are important as a possible carrier of the virus between geographical regions and in acting as a reservoir species. Quail have also been associated with acting as a host to allow re-assortment and viral evolution. Finally swine are important because of the over-lapping glycosylation site specificity of swine and human viruses, which suggests that they may act as an intermediate for transmission to humans (Guo et al., 2005).

There are problems in interpreting the quail trees because of the absence of the Shantou sequences from the nucleotide tree because of partial sequencing (Fig. 8). Shantou was the main geographical location for the outbreak of H9N2 in quail between 2000 and 2005 (Xu et al., 2007a; Xu et al., 2007b). Live quail were banned from Chinese wet markets after a study had shown the link to the virus. In the literature it was found that by 2003 this had resulted in the G1-like lineage disappearing from China but the data here show that it was still present in quail in the Shantou region until 2005 (Choi et al., 2004). The trees show that quail are hosts for all of the main lineages clade A, clade B (G1-like lineage) and the main Chinese clade. This would support the theory that quail have been the host species responsible for the diversification of the H9N2 sub-type, especially given that the original sequence of G-like lineage that is the prototype sequence for clade B was originally found in a Hong Kong quail (Hossain, Hickman & Perez, 2008). This also fits with the epochal model of viral evolution proposed by Wagner (2008). There are periods of neutral drift, which are punctuated by selection events. Here the selection event is the formation of a new lineage within a different host species after a long period of neutral drift within clade A.

Figure 8 The qual nucleotide and amino acid phylogenetic trees.

(A) The nucleotide phylogenetic tree, (B) The amino acid phylogenetic tree. Numbers on the internal branches are the bootstrap values.

Like quail ducks provide a host for the clade A and main Chinese clade viral lineages (Fig. 9). There is only a single example of a clade B sequence in ducks, and so they might not be very effective carriers of this virus or this might be explained by their geographical exposure to that lineage (Perez et al., 2003). Ducks are of concern as a host species as they can contribute to the spread of this lineage over a wider geographical region. In the global phylogenetic tree sequences from ducks are often found clustered with those from quail showing that there is frequent transfer between the two hosts.

Figure 9 The duck nucleotide phylogenetic tree.

Numbers on the internal branches are the bootstrap values.

Most of the swine cases have been within the main Chinese clade of sequences (Fig. 10). There is a single sequence from Korea in 2004 that belongs to clade A. This is important in guiding how we monitor disease outbreaks because it shows that the most geographically widely distributed clade can also produce infection in pigs. The swine flu epidemic of H1N1 originated in Mexico whereas China had been the main focus of surveillance, because of the frequency and previous circulation of the virus. This is also true of H9N2 monitoring which is focused on Southeast Asia, but shows that more widespread monitoring is important in the early detection of pandemics.

Figure 10 The swine amino acid phylogenetic tree.

There is no obvious geographical or temporal pattern in infection in pigs and there are multiple transmissions between birds and pigs that produce a tree with many different sub-clades usually with only a small number of members. This is consistent with there not being a widespread circulation of the virus within pigs, which would give a larger homogeneous cluster of swine viruses from different times and locations due to pig to pig transmission. Previously five amino acids changes had been identified as being swine specific, but none of these were conserved within the swine sequences or even within a single swine clade (Xu et al., 2004). There are a few cases of the S145N mutation that has been identified to change the antibody epitope but this again is sporadic (Ping et al., 2008).

Identifying the amino acid changes responsible for clade formation

The X-ray crystal structures are available for the H9 hemagglutinin and so it is possible to map the amino acid changes responsible for differentiating between the different clades onto the structure (Ha et al., 2001; Ha et al., 2002). The region between amino acids 128 and 275 makes up the receptor subdomain. This domain is responsible for binding to the cellular membrane as part of viral invasion. The stem domain is made up of the first 60 amino acids of the N-terminus and the final 275 amino acids in the C-terminus of which the last 221 are cleaved by proteolysis of the precursor protein at a conserved arginine to produce a second peptide chain. In between these domains is the remains of a catalytic domain—the vestigial enzyme domain (Ha et al., 2002). The amino acids that are specific to the four most distinct clades are given in Table 1.

Table 1 The amino acid differences between clades A and B with respect to the main Chinese clade.

Numbering is for the complete H9 hemagglutinin protein. Changes are from the conserved sequence (main Chinese clade) to the clade specific sequence.

Clade A	Clade B	
V4 → T	V3 → T	
T8 → A	V15 → T or A	
T38 → I	N40 → T	
A47 → T	L107 → T	
I79 → V	S121 → T	
R92 → K	S143→ T	
L122 → F	S147 → T	
S127 → N	S158 → N	
Q164 → H	N167 → G	
R180 → E	A168 → L	
T206 → A	M187 → V	
R335 → A	N191 → H	
S337 → D	T204 → I	
K381 → E	R205 → N	
V394 → I	I217 → L	
K473 → N	D239 → N	
N496 → D	R294 → K	
	T299 → S	
	V306 → I	
	N313 → T	
	V318 → I	
	V333 → I	
	S353→ P	
	I429 → V	
	V469 → M	
	I537 → L	

The presence of four key amino acids has been shown to be essential for droplet transmission of the virus H183, A189, E190 and L226 that correspond to residues H191, A197, E198, and L234 in the H9 numbering (Sorrell et al., 2009).

There are 17 amino acids that distinguish between Clade A and the consensus sequence, only three of these are in the receptor domain and so the majority are in the stem domains. Three mutations are in the enzymatic domain of which the most significant of which is the replacement of a serine at residue 127 with an asparagine as this is on the boundary of the domain and this creates another potential glycosylation site. The receptor domain changes are Q164H, R180E and T206A. Of these the replacement of the basic arginine group by an acidic glutamic acid is the most interesting, because of the change in polarity, none of the amino acid changes affected either the glycosylation sites or altered the residues that were identified as key to viral droplet transmission. Only T8A had been previously identified as an important change when it was shown to be involved in host specificity (Perez et al., 2003).

There are 26 amino acid changes that distinguish Clade B. Eleven of these changes can be found in the receptor subdomain. Many of the changes are between leucine, isoleucine and valine. These are conservative changes that preserve the hydrophobicity of the amino acid but change the steric interactions. Another significant proportion of the changes is substitutions to threonine from serine or valine. Three of the serine to threonine changes occur in the receptor domain and this might reflect an altered binding affinity for a larger binding partner. Position A168 had been identified previously as under positive selection (Fusaro et al., 2011). This is supported by the mutation to leucine in this clade. Of the key amino acid changes required for droplet infection only the N191H mutation is clade specific.

The amino acid changes responsible for differentiating between clades gives some support to previous studies that have tried to identify residues that are under selection (Fusaro et al., 2011). However most of the clade specific changes have not been previously identified in the literature on the evolution of antigenicity (Kaverin et al., 2004; Skehel & Wiley, 2000). The amino acids responsible for glycosylation are conserved throughout the phylogenetic trees, although some of the amino acid changes introduce new asparagine residues, which could be new sites for glycosylation (Guo et al., 2000; Zhang et al., 2004). There are a very large number of sequences with the L234 substitution required for droplet infection but these are not specific to any of the major clades and this shows that the mutation has arisen multiple times.

Conclusions

The sequence databases are growing at a hyperbolic rate, and with next generation sequencing, this level of growth is likely to continue for the foreseeable future. There are two challenges for dealing with this data. The first is computational that requires improved algorithms and implementations especially as we are moving to more computationally intensive methods of analysis. The second is the quality of the data collection itself. Currently data collection is not systematic and this seriously affects the reliability of the models that can be built. Sampling is badly skewed to certain geographical locations, China being a prime example, while others are ignored (Africa and South America). Surveillance of wild birds is particularly problematic. Where there have been international efforts such as in the European Union study of the spread of H5N1 in wild birds, even this was incomplete with no data from Spain and Eastern Europe, and only partial data for France and Germany (Hesterberg et al., 2009).

There are also problems with partial and truncated sequences, as these often have to be excluded from analysis. This is becoming less of a problem as sequencing methods become cheaper and so complete sequences become more widely available.

Another area where there needs to be a significant improvement is in the quality of the sequence annotations in the databases. For effective phylogeographic analysis it is important that future data should be annotated with as much geographic data as is possible, this must include GPS coordinates and further GIS (geographic information system) information to include habitat and urbanisation measurements would be ideal (Scotch et al., 2011; Yasué et al., 2006).

In this paper there are clearly three different principal clades; Clade A—Wisconsin like, Clade B—G1-like and the Main Chinese Cluster—Y280-like, but it is not clear when a new lineage has arisen and when they are no longer “like” the prototype sequences. At the sub-lineage level the assignment of cluster and clades strongly depends on the sampling of the sequences and this has produced some conflicts between different assignments in the literature. Geography is a much stronger determinant of lineage rather than the avian host, which seems to provide a much weaker barrier to spread of the virus. However there is a definite barrier between bird species and pigs as hosts.

The clade analysis has provided insight into the functional and structural evolution of the protein. There is only a limited overlap between the residues identified in this study as important in clade differentiation and those identified as significant in the existing literature. There is therefore a need for further investigation of the functionality of these newly identified amino acid changes.

Supplemental Information

Figure S1 Expanded phylogenetic tree

In this figure all of the clades have been expanded to show the leaf nodes.

Click here for additional data file.

Figure S2 Alignment of the unusual sequence to the prototypes of the three H9N2 hemagglutinin lineages

This is an alignment produced using clustalw for the nucleotide sequences of the Shandong sequence from 1999 to the prototype sequences from the three previously determined lineages.

Click here for additional data file.

Table S1 Model test results for the evaluation of the different amino acid substitution models used to build the phylogenetic trees

Results are evaluated using Bayesian Information Criteria, Akaike’s Information Criteria and Log Likelihood.

Click here for additional data file.

Table S2 Model test results for the evaluation of the different nucleotide substitution models used to build the phylogenetic trees

Results are evaluated using Bayesian Information Criteria, Akaike’s Information Criteria and Log Likelihood.

Click here for additional data file.

Supplemental Information 5 This contains the raw protein and nucleotide sequence files in fasta format

There are two directories one for the protein sequences and a second for the nucleotide sequences. There are also two MEGA files that include the nucleotide alignment (.mas file) and the phylogenetic trees (.mts file).

Click here for additional data file.

AD would like to thank Dr Lorna Tinworth for her careful reading of the manuscript, the four anonymous referees for their helpful comments on the current submission, and Prof. Stronzo Bestiale for his comments on an earlier draft of the manuscript.

Additional Information and Declarations

Competing Interests

Author Contributions

The authors declare there are no competing interests.

Andrew R. Dalby conceived and designed the experiments, performed the experiments, analyzed the data, wrote the paper, prepared figures and/or tables, reviewed drafts of the paper.

Munir Iqbal conceived and designed the experiments, wrote the paper, reviewed drafts of the paper.

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
