# Peer review of "A global phylogenetic analysis in order to determine the host species and geography dependent features present in the evolution of avian H9N2 influenza hemagglutinin"

_PeerJ, doi:10.7717/peerj.655_

## Round 0.1 · original submission · Major Revisions

Note the reviewers have a number of issues for you to consider in a revised draft of your paper. Importantly, Reviewer 1 identifies another data set that would need to be incorporated into your paper to keep it up to date and most informative. Reviewers 2&3 also mention additional studies for consideration. There are also a number of issues that need attending. However, all the reviewers felt your paper informative, interesting, and appropriate for the journal. I look forward to your revised draft.

Reviewer 1 ·

Basic reporting

The manuscript claims to describe the most recent evaluation of AIV H9N2 worldwide, and cites two publications of Fusaro et al., 2011 and Monne et al., 2013. However a more recent publication analyses athe most comprehensive collection of H9N2 isolates from the Middle East and Israel, and should be included in the body of information of the present study:

Davidson I., A. Fusaro, A. Heidari, I. Monne and G. Cattoli (2014). Molecular evolution of H9N2 avian influenza viruses in Israel. Virus Genes, 48: 457-463
.

Experimental design

no comments

Validity of the findings

Until latest data is included the manuscript is incomplete

Reviewer 2 ·

Basic reporting

No Comments

Experimental design

Introduction:
Line 69: the authors should be more specific in stating the research question, this will highlight the relevance and major contributions of their study. Consider “the current study presents a detailed or comprehensive phylogenetic analysis that includes sequences from all different geographical regions where H9N2 have been reported up to 2013…” or perhaps state the hypothesis that host species and geographic distribution have shaped the evolution of H9 HA and that this is reflected in the topology of the phylogenetic relationships between the nucleotide and protein sequences.

Materials and methods:
The methods are well described and the authors used Maximum Likelihood to generate the phylograms. All trees in the figures are presented as cladograms, omitting the information of genetic distances between different taxa. While, this facilitates to see the order of the clades and the overall tree topology, very important information is missed, and could be included as branch labels in the trees. It should be clearly stated in the methods that the cladograms were used to do the interpretation and clarify why such decision was made. In addition, phylograms could be used in the final figures including a scale bar for branch lengths.

Validity of the findings

The findings presented here are interesting as they present a very detailed distribution of viruses from different hosts and geographic locations. The authors reveled interesting amino acid mutations responsible of clade diversification that could be used in future studies to analyze phenotype differences between viruses from different clades, as they spread or jump between species.

Comments on specific sections:
Results and discussion
Line 119: separate section title from previous paragraph.
Line 120: consider comparing clades to those defined by Dong et al. as many of the sequences are shared between these studies.
Line 121: The first statement is unclear, the idea should be completed or removed.
Line 165: Provide the method of phylogenetic inference (distance of ML) was used by Kim et al and explain why this difference is relevant in this context.
Line 166 to 169: In reference to previous statement, in addition to bootstrap support, testing with other methods of phylogenetic inference one same data set is useful to confirm tree topologies, if this was not done it may be worth mentioning that this could be a way to further confirm the topology that resembles the “correct tree”. In line 167, define state which bootstrap values were considered high enough (>80%, >90%) to provide credibility to different clades.
Line 200: replace “Swine” with “swine”.
Line 203: this sentence is redundant, consider “this hypothesis cannot be confirmed without a longer period of absence of detections of viruses from this clade”.
Line 210: Looking at the branch lengths in the phylograms between the Shandong sequence from 1999 and those from 2003 in sub-clade 8 of the main Chinese lineage could help to resolve whether this sequence is an outlier, or only reflects poor taxon representation over time.
Line 287: Surveillance hotspots are defined based on the frequency and previous circulation of viruses in specific populations (e.g. Southeast Asia), however, this does not prevent that viruses may emerge in other places. The fact that the H1N1 pandemic originated in Mexico, only represents an independent event from a geographical region where surveillance was missing. The statement “It was a surprise…” is subjective and irrelevant, should be changed or removed.
Line 291: “transmission between birds” an pigs suggest two-way transmissions, rather that multiple independent introduction from birds to pigs. Please clarify this statement.

Additional comments

The manuscript by Dalby and Iqbal entitled: “A global phylogenetic analysis in order to determine the host species and geography dependent feature present in the evolution of avian H8N2 Influenza Hemagglutinin” describes a comprehensive analysis of different phylogenetic clades identified for H9 hemmaglutinin nucleotide and protein sequences available from all over the world. A detailed analysis was done at the level of sub-clades to investigate the diversification of the H9 viruses between host species and geographic locations. Similar analyses have been done for H9N2 viruses, as described in the introduction section of this paper. The authors mention that most of the studies have been restricted to specific geographical regions, however, an additional study conducted by Dong et al 2011. PLoS ONE 6(2): e17212 should be mentioned, in which H9N2 virus sequences from different continents, collected between 1966 and 2009, where analyzed to define genotypes and lineages. As sequence data continue to accumulate, the study by Dalby and Iqbal is of relevance as it presents an updated analysis. Its findings should be published, after some minor revisions.

Figure legend for Figure 8. Typographical error, substitute "qual" by "quail".

Reviewer 3 ·

Basic reporting

Figures are not properly labelled. In particular, the bootstrap values should be added in figures 1 to 6, in figure 10 and in the phylogenetic trees in the supplemental materials. In addition, the clade designation should be shown in the phylogenetic trees constructed for each geographical and host subset. Moreover, all the phylogenies are displayed as cladograms, which exhibit only the phylogenetic relationship among the taxa and the data set. In this kind of study the authors should use phylograms where the branch lengths are drawn proportionally to evolutionary distances. Also the scale bar of nucleotide substitution per site should be displayed.

Experimental design

Dong et al., PlosOne, 2011 have already described most of the findings reported in this pape. Dong et al et al. analysed the complete genome of 571 H9N2 viruses collected between 1966 and 2009 and identified 7 lineages in the HA phylogeny. They also observed that “the H9N2 lineages had marked host and geographical differences in phylogeny” as suggested in the paper under review. Although, Dalby and Iqbal included much more sequences (of just the HA gene) in their manuscript, the findings seem very similar. The authors should at least cite this manuscript and take into account the proposed nomenclature system.
It is not clear the criteria used for the selection of the datasets. The authors decided to include partial sequences in the protein dataset, removing only those with truncations >40 aa. Differently, for the nucleotide dataset they retrieved only complete sequences. It is not clear why the authors used these two different criteria for the selection of the datasets. I would suggest the use of the same sample dataset for both nucleotide and protein analysis. Indeed, the different number of sequences and the different HA regions analysed may have some impact on the topology of the phylogenetic trees, making any comparison difficult.
In addition, the authors do not specify the criteria used to define the different clades and subclades (i.e. bootstrap values, average distance between and within clades). As the entire manuscript is constructed around the description of these clades, it is essential to define how these were designated. Moreover, displaying the ML phylogenetic trees as cladograms the authors loose information about the amount of evolutionary changes. I suggest the use of phylograms, which show the degree of evolutionary changes besides the tree topology. It is not clear why the authors assigned new names to each clade and did not use the nomenclature used in previous papers (such as Dong et al., PlosOne, 2011). This will avoid any further confusion in the H9N2 classification.
The trees obtained for the clade analysis of the geographical and host subsets take the samples of each subset off their evolutionary context, making difficult to locate the taxa in the complete H9 phylogeny. I would suggest the authors mark the different subsets in the complete phylogenetic trees, for example colouring the branches according to the different host or location. In this way the presence of geographic- and host-clustering would be clearly and immediately identifiable and will allow determining i) the genetic relationship between viruses from different host or geographic regions, ii) the virus jump between different species and iii) their diffusion among identified regions. To make the phylogenies fitting a page size within the main manuscript the tip labels can be removed. A program like FigTree (http://tree.bio.ed.ac.uk/software/figtree/) could help the authors to provide a better graphic visualization of their trees.

Validity of the findings

In the conclusions (line 365-366) the authors stated “there is a definite barrier between bird species and pigs as hosts”. However, in the results they showed the occurrence of multiple transmissions between birds and pigs. The authors should explain better what they mean with the term “barrier”.

Additional comments

Line 18. The work of Sorrel et al. is for H9N2 and not H5N1. Please modify the sentence accordingly
Lines 47-59. This part is superfluous and not very clear (in particular lines 50-54). I would suggest the authors to remove it.
Line 70: The authors should specify that the analysis concern only the H9N2 viruses.
Lines 79 and 81: It is not clear why the authors did not use the GenBank Influenza virus Resource for sequences selection and download and why they used different length criteria for the nucleotide and protein sequences research.
Line 96: Analyses of species trees created for chickens and pheasants were not shown. It is not clear why the authors performed such analyses if they did not show them.
Line 115. The authors should try to perform a bootstrap analysis with at least 100 replicates.
Line 117-118. Please, explain what “conserved sequence” means
Line 121. The sentence “A complete (…) dataset” is not complete. Please modify it.
Lines 142-149: As the outlier sequence (Shandong 1999) belonging to clade 8 is the only sequence from 1999 of this clade it could be a sequence obtained from a more recent virus assigned to the wrong sample. The authors should consider also this hypothesis.
Line 167. Unfortunately, the bootstrap values are not displayed in figure 2.
Line 190. Please, explain what do you mean with the sentence “clade C fall between clade A and B”. Is the clade C derived from clade A and the progenitor of clade B?
Line 205: It is not clear if these “new clades” had been previously identified as belonging to Beijing/1/94 lineage or if they are described here for the fist time.
Lines 2019-224: It is not clear why Korea, Iran and Israel are considered more interesting than the other states.
Lines 236-237: The impact of vaccination on virus evolution is very difficult to assess from the phylogenetic trees displayed in this paper.
Lines 238-241: It is not clear the correspondence between clades 1 to 7 visualized in figure 7 and the clade A, B, C and the Chinese Clade. They are probably subclades identified within the main clades. To avoid confusion, the authors should show the main clades to which these subclades belong in the phylogenetic tree. In addition, I suggest the authors name them as “subclades” or “group”. As previously mentioned, the authors should also define the clade definition criteria for the identification of the main clades and subclades (i.e. bootstap values, within and between-group nucleotide/amino acid distance). The sentence “This is why this clade (…) sequence” is not clear. Are the authors suggesting that this clade is the progenitor of the other clades circulating in Iran? Why the “deepest origin” of this group should be explained by the fact that it contains duck sequences? Please explain.
Lines 270-272. Are quails the only specie that is “host for all the main lineage”? What about chickens?
Line 282: Please specify how many times the transfer between ducks and quails has been observed.
Lines 284-289. The concept expressed in this paragraph is not clear. Please, explain it better.
Line 306. In table 1 only mutations between clade A and B and the conserved sequence are reported. Please change the table or the sentence accordingly. Please, explain what “specific” means (are the mutations present in 100% of the sequences within each analysed clade?) and define the “conserved sequence”. Instead of comparing sequences against a conserved sequence, I suggest the authors identify all the mutations that are unique of each clade (A, B, C, main Chinese clade).
Line 312. Please replace “domains” with “mutations”.
Line 316. The authors should better explain what they intend with the sentence “most interesting”.
Line 320. These 26 changes distinguish only clade B from the “conserved sequence”. Please specify this better.
Lines 328-329 and 335-336. This is just a pure speculation. Please delete this sentence.
Figure 8. Please replace “qual” with “quail”.

Reviewer 4 ·

Basic reporting

Generally, it is a quite informative and well written reviewing manuscript.
The author analyzed the huge amount of sequence data from NCBI and categorized the HA proteins of H9N2 into clade A to main Chinese clade. Base on complete phylogenetic analysis of the H9N2 hemagglutinin sequences, the author built a picture of the geographical and host specific evolution of the H9 hemagglutinin proteins.

Experimental design

No Comments

Validity of the findings

There are some concerns that need to be clarified prior to publication in PeerJ.
Comments:
1. Clade A covered to many divergent virus strains. In this reviewer's opinions, North American lineages, Y439, Korea, and Eurasian wild bird lineages of clade A might be more distinctive each other rather than clade C and B based on phylogenic distance (fig. 2).
Please clarify how these distinctive lineages could be clustered into one Clade (A).
2. There were too many separated trees. The authors made separate polygenetic trees of quail, duck and swine (Fig. 8 to 10). However, if these trees are merged into one tree, it is more easy to understand the host species determinants and each evolution patterns.
3. Table 1 could be more informative if it is translated as a figure with HA domain diagram. Mapping the distinctive changes associated to the proposed clades would perhaps provide a better picture on how these different clades could be distinguished, like what happened to HPAI H5 viruses.

Additional comments

Generally, it is a quite informative and well written reviewing manuscript.
The author analyzed the huge amount of sequence data from NCBI and categorized the HA proteins of H9N2 into clade A to main Chinese clade. Base on complete phylogenetic analysis of the H9N2 hemagglutinin sequences, the author built a picture of the geographical and host specific evolution of the H9 hemagglutinin proteins.

However, there are some concerns that need to be clarified prior to publication in PeerJ.
Comments:
1. Clade A covered to many divergent virus strains. In this reviewer's opinions, North American lineages, Y439, Korea, and Eurasian wild bird lineages of clade A might be more distinctive each other rather than clade C and B based on phylogenic distance (fig. 2).
Please clarify how these distinctive lineages could be clustered into one Clade (A).
2. There were too many separated trees. The authors made separate polygenetic trees of quail, duck and swine (Fig. 8 to 10). However, if these trees are merged into one tree, it is more easy to understand the host species determinants and each evolution patterns.
3. Table 1 could be more informative if it is translated as a figure with HA domain diagram. Mapping the distinctive changes associated to the proposed clades would perhaps provide a better picture on how these different clades could be distinguished, like what happened to HPAI H5 viruses.

Overall, it quite well analyzed and informative reviewing manuscript.

---

## Round 0.2 · Minor Revisions

Just a bit more clean up and you will be there. Please note these final comments, do a thorough re-read of the paper for typos, and you'll be good. Thanks.

Reviewer 2 ·

Basic reporting

The manuscript "A global phylogenetic analysis in order to determine the host species and geography dependent features present in the evolution of avian H9N2 Influenza Hemagglutinin" by Dalby and Iqbal, was modified accordingly to include most of the comments suggested by the reviewers.

Experimental design

Authors appropriately explained the rationale of using a cladistics approach.
Line 132 (revised manuscript): The authors explain to the reviewers why species specific trees were created for different hosts, however this explanation was not included in the manuscript. This approach has been previously used by Worobey et al, 2014 (Nature 508, 254–257), however host-species specific phylogenetic analysis has not been common practice in the field of Influenza, it would be important to emphasize this to clarify this section.

Validity of the findings

Main points of discussion (cladistic vs phylogenetic analysis) were clarified, and bootstrap values were added to the trees where appropriate. The studies by Davison et al 2014 and Dong et al 2011 were included, and the results were discussed in the context of recent data, improving the quality of the manuscript.

Additional comments

Substitute the word "paper" by "study" to refer to this and other studies done by other groups.

---

## Round 0.3 · accepted · Accept

Thanks for the quick and useful turn around. I think you are good to go now.